# Theoretical and Experimental Investigation of Warpage Evolution of Flip Chip Package on Packaging during Fabrication

**DOI:** 10.3390/ma14174816

**Published:** 2021-08-25

**Authors:** Hsien-Chie Cheng, Ling-Ching Tai, Yan-Cheng Liu

**Affiliations:** 1Department of Aerospace and Systems Engineering, Feng Chia University, Taichung 407, Taiwan; she850303@yahoo.com.tw (L.-C.T.); P0700100@o365.fcu.edu.tw (Y.-C.L.); 2Ph.D. Program of Mechanical and Aeronautical Engineering, Feng Chia University, Taichung 407, Taiwan

**Keywords:** flip chip package on package, finite element analysis, viscoelastic behavior, process-induced warpage, trace mapping, effective modeling

## Abstract

This study attempts to investigate the warpage behavior of a flip chip package-on-package (FCPoP) assembly during fabrication process. A process simulation framework that integrates thermal and mechanical finite element analysis (FEA), effective modeling and ANSYS element death-birth technique is introduced for effectively predicting the process-induced warpage. The mechanical FEA takes into account the viscoelastic behavior and cure shrinkage of the epoxy molding compound. In order to enhance the computational and modeling efficiency and retain the prediction accuracy at the same time, this study proposes a novel effective approach that combines the trace mapping method, rule of mixture and FEA to estimate the effective orthotropic elastic properties of the coreless substrate and core interposer. The study begins with experimental measurement of the temperature-dependent elastic and viscoelastic properties of the components in the assembly, followed by the prediction of the effective elastic properties of the orthotropic interposer and substrate. The predicted effective results are compared against the results of the ROM/analytical estimate and the FEA-based effective approach. Moreover, the warpages obtained from the proposed process simulation framework are validated by the in-line measurement data, and good agreement is presented. Finally, key factors that may influence process-induced warpage are examined via parametric analysis.

## 1. Introduction

In recent years, there has been explosive and continuous growth in the consumer market for various smart products and Internet of Things (IoT) products, as well as the developing requirements of 5G communication, artificial intelligence (AI), and autonomous vehicles. Advanced packaging technology like flip chip packaging [1], wafer level packaging [2], and flip chip chip-scale packaging (FCCSP) [3,4] was introduced to achieve high I/O density, excellent electrical performance and miniaturization, and thus is commonly used in high-end smart chips in recent years. However, the physical limitations of Moore's law [5] make it difficult for electronic packaging to continuously shrink and functionally improve, prompting the development of new packaging technologies. Among the many solutions, the system-in-package (SiP) for heterogeneous integration is the current alternative, and is one of the most feasible methods for “More than Moore” or even “Beyond CMOS”.

SiP technology may have a two-dimensional (2D) planar configuration, a three-dimensional (3D) vertical stacking configuration, or an integrated (hybrid) configuration. 3D packaging technology can be categorized into package stacking, like package-on-package (PoP) and package-in-package (PiP), wire-bonding [6], and through silicon via (TSV)-based 3D IC stacking [7]. In addition to high I/O quantity and miniaturization, further requirements of multi-functionality have aroused the development of the flip chip package-on-package (FCPoP) technology. This packaging technology has attracted a great deal of attention from the semiconductor packaging industry due to its compelling features including heterogeneous integration capability, high electrical performance, high bandwidth, low power consumption, small form factor, low cost, etc., leading to wide potential applications, such as high-performance application CPUs. To date, research on FCPoP has been exceptionally limited. Among the limited literature, the focus was placed on packaging construction [8,9]. For example, Hsieh et al. [9] proposed a PoP technology with the flip chip structure for mobile device applications, which can be a bare die PoP packaging technology, a molded laser PoP packaging technology, or a silicon interposer substrate PoP technology to achieve thickness and warpage reduction.

Reliability and yield are the two most important issues in microelectronics assembly [2]. Process-induced warpage during fabrication is one of the root causes for the poor assembly reliability and yield [10,11,12] Although the bottom layer structure of the FCPoP assembly employs a mature flip chip packaging technology, it is still indispensable to stack an interposer as a bridge to connect to the memory, potentially causing a more serious mismatch of the coefficients of thermal expansion (CTE) of the different materials. The excessive CTE mismatch together with high process temperatures may induce considerable residual stress/strain, which will generate not only serious warpage but also throughput loss. Therefore, to effectively master and control the process-induced deformations of the assembly is the key to the success of the technology. Compared with FCPoP, there have been many studies in the past on the warpage behavior of FCCSPs [3,12]. It was found that viscoelastic behavior of the molded underfill material would contribute to their warpage performance. In addition, the substrate or interposer is also a crucial factor dominating the warpage behavior. To suppress the warpage of packages, substrates with a core material have been widely used. As portable devices become thinner, so do the package size and substrate, coreless substrates have recently become increasingly popular in electronic packaging [3]. However, a lack of rigid core material for structural support may cause the warpage to be even more sensitive and pronounced.

This research aimed to establish a process simulation framework for predicting the warpage behavior of an FCPoP assembly during fabrication. The viscoelastic behavior and volumetric shrinkage of the epoxy molding compound (EMC) [11,12] were included in the process modeling. The FCPoP consisted of one bottom orthotropic coreless substrate and one top orthotropic core interposer, each of which comprised several copper (Cu) circuit layers of multi-material and multi-scale structures and complex geometric features. These layers may have a significant influence on process-induced warpage because of the high modulus and CTE of Cu. Thus, in order to offer an accurate prediction, these Cu circuitries need to be accurately modeled in the modeling. However, because of their high geometric and structural complexity, efficiently and thoroughly modeling, these Cu circuit layers presents great challenges. In order to greatly improve the computational and modeling efficiency while accommodating the need for good prediction accuracy, a novel effective approach was proposed to effectively simulate the global thermo-mechanical behavior of the orthotropic coreless substrate and core interposer. The effectiveness of the proposed effective approach was demonstrated by comparing the predicted effective elastic properties with the results of other effective approaches. The proposed process simulation model was validated using in-line warpage measurement data. Finally, parametric analysis is performed to assess the influence of several material and geometry parameters on the process-induced warpage of the FCPoP.

## 2. Structure and Fabrication Process of FCPoP

The research vehicle was an FCPoP assembly, as shown in Figure 1, that was primarily composed of an FCCSP package, an EMC, a core interposer, and Cu core solder balls (CCSBs). The main structure of the FCCSP package included a silicon chip, an underfill, Cu pillar bumps, and a coreless substrate. The FCCSP package usually is used for high-end processors. In addition, the core interposer was applied to facilitate the connection with the HBM for heterogeneous integration, the CCSBs were utilized to connect the bottom FCCSP and the core interposer, and the EMC was used to protect the solder balls. To minimize the package profile, a three-layer 100 μm embedded trace substrate (ETS) was applied, as schematically depicted in Figure 2, which mainly included two solder mask (SM) protective layers, two prepreg (PP) dielectric layers, and three metal (Cu) layers, with the circuitries filled with either SM or PP material. The chip was 9.36 mm in length, 8.76 mm in width, and 70 μm thick. The chip was connected to the coreless substrate using 2500 Cu pillar bumps, which were 40 μm in length, 70 μm in width and 58 μm in height. The gap between the chip and coreless substrate was filled with an underfill via the capillary action. The top layer was stacked on a 90 μm thick two-layer core interposer. In the FCPoP assembly, there were a total of 550 CCSBs with a diameter of 190 μm. Finally, the EMC was filled between the substrate and interposer to form an FCPoP assembly with a length and width of 14 mm and a thickness of 500 μm. Figure 3 describes the main fabrication process steps; i.e., the die bonding process (steps 1–3), underfill cure process (steps 3–6), interposer bonding process (steps 6–9), mold cure process (steps 9–12) and temperature elevation process (steps 12–13), and also the corresponding process temperatures.

## 3. Theoretical Models

### 3.1. Linear Viscoelasticity 

The properties of viscoelastic materials can be divided into elastic and viscoelastic parts. The elastic part can react immediately and obey Hooke's law when subjected to a fixed load. Conversely, the viscoelastic part gradually increases the strain (creep) or reduces the stress (relaxation), and eventually reaches stability. Polymer materials typically show viscoelastic relaxation behavior [13,14,15,16]. The stress relaxation effect is usually dominated by chemical phenomena at high temperatures for a long period of time. To describe the viscoelastic relaxation behavior, the generalized Maxwell model is most commonly used. It is composed of several Maxwell elements and an independent spring combined in parallel. The time-dependent viscoelastic stress relaxation modulus can be expressed by a Prony series mathematical representation [11]:(1)E(t)=∑k=1nEkexp(−tςk)+E∞
where *n* is the number of Maxwell elements, *E_k_* denotes the modulus of each Maxwell element, *E*_∞_ represents the relaxed modulus, $\varsigma_{k}$ stands for the relaxation time, and *t* is the time. Furthermore, the modulus of each Maxwell element can be described as:(2)$$E_{k}=c_{k}E_{0}$$
where $c_{k}$ denotes the weighting factor and $E_{0}$ represents the unrelaxed modulus, expressed as
(3)$$E_{0}=\sum_{k=1}^{n}E_{k}+E_{\infty }$$

Combining Equations (1)–(3) yields: (4)$$E(t)=E_{0}\left[c_{\infty }+\sum_{k=1}^{n}c_{k}\exp\left(-\frac{t}{\varsigma_{k}}\right)\right]$$

The properties of polymer materials, in either a glass or a rubbery state, show a strong relationship with temperature. The glassy state refers to the polymer material at a temperature higher than the glass transition temperature (*T_g_*). In contrast, the rubbery state refers to the polymer material at a temperature lower than *T_g_*. The Young's modulus and CTE have a large variation during the phase transition state. The time-temperature superposition (TTS) principle is often applied to depict the time-temperature dependence of the linear viscoelastic behavior. The TTS principle illustrates that the relaxation curve of the material's modulus and time (or frequency) at a certain temperature is analogous to the relaxation curve of the adjacent temperature. The relaxation curve at each temperature, except the reference temperature, is translated in the logarithmic time domain to form the relaxation curve of the material at the reference temperature. The value of this translation highly depends on the temperature, the reference temperature and the properties of the polymer materials. The TTS principle can be simply expressed as follows:(5)$$E\left(t,T\right)=E\left(\tau ,T_{0}\right)$$
where τ stands for the reduced time when $T_{0}$ < T, which can be derived below:(6)$$\tau =\int t\kappa_{T}\left(t\right)dt$$

In Equation (6), $\kappa_{T}$ is the shift factor of temperature. This parameter can be approximated by the well-known Williams-Landel-Ferry (WLF) model [17] as:(7)$$\log_{10}\kappa_{T}=\frac{-a_{1}\left(T-T_{ref}\right)}{a_{2}+\left(T-T_{ref}\right)}$$
where *T_ref_* is the reference temperature, and *a*_1_ and *a*_2_ represent the coefficients of the curve fit. Basically, they are highly dependent on the materials and the reference temperature.

### 3.2. Linear Elastic Mechanics 

Polymer materials, such as EMCs, will exhibit volumetric changes or chemical shrinkage during the mold cure process due to chemical reactions. In essence, the extent of volumetric change highly depends on the material’s cure state in an isothermal isobaric ensemble. Consider that a polymer cube has a length of one-unit side before curing. The volumetric change ($\tilde{V}$) of the unit cube after full curing is written as:(8)$$\tilde{V}\approx 1-3\Delta l$$
where Δ*l* is the variation of the unit side length. The corresponding strain due to the volumetric change $\varepsilon_{vc}$ can be expressed as:(9)$$\varepsilon_{vc}=\frac{1}{3}\tilde{V}$$

The total strain of the polymer materials is the sum of the volumetric change-inducted strain ($\varepsilon_{vc}$), elastic strain ($\varepsilon_{e}$), and thermal strain ($\varepsilon_{thermal}$). According to Hooke’s law, the relationship between the elastic strain and stress can be written as follows:(10)$$\left\{\varepsilon_{e}\right\}=\left[B\right]\left\{w\right\}-\left\{\varepsilon_{vc}\right\}-\left\{\varepsilon_{thermal}\right\}$$
where {w} is the nodal displacement vector and [B] represents the strain-displacement matrix.

## 4. Numerical Modeling and Material Characterization

### 4.1. Effective Modeling 

#### 4.1.1. The Proposed Effective Method

To effectively capture the process-induced warpage behavior of the FCPoP assembly relies on a dependable and accurate thermo-mechanical characterization of the core interposer and coreless substrate, consisting of several Cu circuit layers with multi-material and multi-scale structures and complex geometric features. Accurately and fully modeling them presents great challenges due to the requiring extensive, tedious effort and cost required. To ease the modeling challenges, the core interposer and coreless substrate were approximated as an equivalent homogeneous medium and their effective elastic properties were evaluated by using an effective approach that made use of the powerful electronic computer-aided design (ECAD) trace mapping (TM) method together with the rule-of-mixture (ROM) technique and finite element analysis (FEA).

The ECAD TM method enables a more efficient and accurate representation of tiny, delicate, complex Cu traces, pads and vias surrounded by the PP dialectic material and the SM protective material on the coreless substrate and core interposer [18]. A flowchart of the ECAD TM method is shown in Figure 4. It is noted that the Cu circuit layers consisted of not only Cu traces, pads and vias, but also SM or PP dielectric material. First of all, based on the ECAD model, a uniform regular background mesh with a significant number of very fine first-order brick elements was established on each Cu circuit layer of the coreless substrate and core interposer. Then, the spatially non-uniformly distributed Cu circuitries in the ECAD model were mapped onto the background mesh to obtain a high resolution (HR) Cartesian Cu circuit map and a finite element (FE) model. Based on the volume ratio of Cu and neighboring materials, such as PP or SM on each brick element of the background mesh, the effective isotropic elastic properties were calculated using an ROM technique, by which a complete material property map of the Cu circuit layer was derived (Figure 5). The benefits of the ECAD TM method were a very uniform regular mesh, flexible mesh density control and close match to the geometry of the Cu circuitries.

As soon as the material property map and 3D FE model of the Cu circuit layer were created, FEA was applied to calculate its effective orthotropic elastic properties. For modeling simplification, the orthotropic Cu circuit layer could be approximated as a transversely isotropic material by averaging the effective in-plane elastic properties and further as an isotropic material by averaging the effective in-plane and out-of-plane elastic properties. The 3D FE models of the mapped Cu circuit layer, PP dielectric and SM layers could be combined together to form an integrated 3D FE model of the substrate and interposer. Finally, the effective orthotropic elastic properties of the substrate and interposer as a whole could be derived using FEA. It is worth mentioning that the final modeling step; i.e., approximation of the substrate and interposer as a homogeneous equivalent continuum, may not be indispensable, since FE modeling of the substrate and interposer could be directly carried out using the integrated 3D FE model. This effective approach is hereinafter termed the TM/FEA effective method.

#### 4.1.2. The ROM/Analytical Estimate

The effective in-plane and out-of-plane CTEs of the Cu circuit layers could be also assessed using an analytical estimate integrated with an ROM method (It is alternatively termed the ROM/analytical estimate). Specifically, the effective in-plane CTE $\alpha_{x,y}^{}$ of the Cu circuit layers was evaluated according to the literature [19] using an energy approach, and the effective out-of-plane CTE $\alpha_{z}^{}$ also was derived based on the work of [19], and also as presented in [7]:(11)$$\alpha_{z}^{}=\frac{E_{1}\alpha_{1}\xi_{1}+E_{2}\alpha_{2}\xi_{2}}{\xi_{1}E_{1}+\xi_{2}E_{2}}$$
(12)$$\alpha_{x,y}^{}=\left(1+\upsilon_{1}\right)\alpha_{1}\xi_{1}+\left(1+\upsilon_{2}\right)\alpha_{2}\xi_{2}-\alpha_{z}^{}\bar{\upsilon }$$
where $E_{1}$($E_{2}$), $\alpha_{1}$($\alpha_{2}$), and $\xi_{1}$($\xi_{2}$) are the Young’s modulus, CTE and volume fraction of the Cu (SM or PP), respectively. In Equation (12), the effective Poisson’s ratio $\bar{\upsilon }$ can be simply approximated using ROM as:(13)$$\bar{\upsilon }=\upsilon_{1}\xi_{1}+\upsilon_{2}\xi_{2}$$
where $\xi_{1}$ and $\xi_{2}$ are the Poisson’s ratio of the Cu and SM or PP respectively. Likewise, the effective in-plane elastic modulus $E_{x,y}^{}$ and out-of-plane elastic modulus $E_{z}^{}$ of the Cu circuit layers can be also estimated as:(14)$$E_{x,y}^{}=\frac{E_{1}E_{2}}{E_{1}\xi_{2}+E_{2}\xi_{1}}$$
(15)$$E_{z}^{}=E_{1}\xi_{1}+E_{2}\xi_{2}$$

#### 4.1.3. The FEA-Based Effective Approach

FEA using a detailed fine mesh model could be directly applied to derive the effective orthotropic elastic properties of the Cu circuit layers. This method can be very effective in accurately grasping the crucial parameters affecting the effective properties but require a very tedious, time-consuming and complex procedure to model and simulate the material models [20]. The method is briefly termed the FEA-based effective approach [7]. The underlying idea behind this approach is that the elastic responses of the homogeneous equivalent continuum should be consistent with those of the original continuum.

The effective CTEs of the Cu circuit layers could be simply calculated based on the strength of the materials,
(16)$$\alpha_{i}^{}={\delta_{i}/\left(\Delta T\right)L}_{i}\;(i=x,y,z)$$
where $\delta_{i}$ is the thermal deformation, $\alpha_{i}^{}(i=x,y,z)$ stands for the effective CTE in the *i*-th direction, ΔT denotes the temperature increment, and $L_{i}$ represents the side length of the Cu circuit layers in the *i*-th direction.

In accordance with the generalized Hooke’s law, the stress-strain relationship of an orthotropic material is expressed as:(17)$$\varepsilon_{xx}=\frac{\sigma_{xx}}{E_{x}}+\frac{\upsilon_{yx}}{E_{y}}\sigma_{yy}+\frac{\upsilon_{zx}}{E_{z}}\sigma_{zz}$$
(18)$$\varepsilon_{yy}=\frac{\upsilon_{xy}}{E_{x}}\sigma_{xx}+\frac{\sigma_{yy}}{E_{y}}+\frac{\upsilon_{zy}}{E_{z}}\sigma_{zz}$$
(19)$$\varepsilon_{zz}=\frac{\upsilon_{xz}}{E_{x}}\sigma_{xx}+\frac{\upsilon_{yz}}{E_{y}}\sigma_{yy}+\frac{\sigma_{zz}}{E_{z}}$$
(20)$$\gamma_{yz}=\frac{\tau_{yz}}{G_{yz}}$$
(21)$$\gamma_{xz}=\frac{\tau_{xz}}{G_{xz}}$$
(22)$$\gamma_{xy}=\frac{\tau_{xy}}{G_{xy}}$$
where $\varepsilon (\varepsilon_{x},\;\varepsilon_{y},\;\varepsilon_{z})$ and $\sigma (\sigma_{x},\;\sigma_{y},\;\sigma_{z})$ are the normal strain and stress, respectively, $\gamma (\gamma_{xy},\;\gamma_{yz},\;\gamma_{xz})$ and $\tau (\tau_{xy},\;\tau_{yz},\;\tau_{xz})$ represent the shear strain and stress, respectively, and $\upsilon (\upsilon_{xy},\;\upsilon_{yz},\upsilon_{xz})$ denotes the Poisson’s ratio. In total, there are nine independent effective elastic constants to be determined for an orthotropic elastic material, which are $E_{x}$, $E_{y}$, $E_{z}$, $\upsilon_{xy}$, $\upsilon_{yz}$, $\upsilon_{xz}$, $G_{xy}$, $G_{yz}$, and $G_{xz}$. These constants can be simply derived based on Equations (17)–(22) through FEAs with a set of different loading and boundary conditions. The rest of the effective elastic constants $\upsilon_{yx}$, $\upsilon_{zy}$, $\upsilon_{zx}$ can be readily derived from the fact that the compliance matrix is symmetric.

### 4.2. Process Modeling

In this study, a process simulation framework that incorporated the proposed TM/FEA effective method, thermal and mechanical FEAs and the element death and birth method in ANSYS was introduced. Due to symmetry, a quarter of the FCPoP assembly was simulated via the proposed process simulation framework. Figure 6 reveals the constructed 3D FE model of the FCPoP assembly, which comprised 193,401 nodes and 185,211 solid elements. It consisted of the main components of the FCPoP assembly, including a coreless interposer, a core substrate, an EMC, a silicon chip, solder balls, Cu pillar bumps, and an underfill. The displacement boundary conditions are set to simulate the symmetry boundary condition, where the out-of-plane displacement of the nodes on the symmetry planes were constrained. In addition, the bottom node on the intersecting line of these two symmetry planes was fixed in the z-direction to prevent rigid body motion.

All the materials in the assembly were assumed to be either linearly elastic and isotropic or orthotropic except the EMC, which was assumed to be linearly viscoelastic. It was noteworthy that the temperature dependence of these materials and the effects of curing shrinkage of the EMC were also taken into account in this investigation. The temperature-dependent elastic properties of the components in the assembly are characterized using a thermal-mechanical analyzer (TMA) (TA Instruments, New Castle, DE, USA) and a dynamic mechanical analyzer (DMA) (TA Instruments, New Castle, DE, USA), as shown in Figure 7.

The stress-free temperature of the EMC was defined as its cure temperature. The curing process of the EMC involved two processes: in-mold cure (IMC) and post-mold cure (PMC). Typically, the IMC process applies a lower temperature and a shorter curing time to increase the stiffness of the EMC. Subsequently, a PMC process with a higher temperature and a longer duration was utilized to completely cure the EMC. However, during the curing, the EMC would experience a volumetric (chemical) expansion or contraction. The measured volumetric change data provided by the manufacturer was applied. It was reported that the EMC during the curing process from the gel point (where the stiffness of the EMC is nearly developed) to a full cure state would cause a 0.09% volumetric shrinkage. The process modeling for the warpage prediction of the FCPoP assembly during fabrication closely adhered to the process steps shown in Figure 3.

## 5. Results and Discussion

### 5.1. Characterization of EMC Viscoelastic Properties

In this work, a DMA measurement system was applied to conduct the stress relaxation experiments in the frequency domain through a three-point bending mode. The storage moduli of the EMC under a 0.5% applied strain over a wide frequency scan ranging from 0.1 Hz to 100 Hz and a broad isothermal temperature range of 25–260 °C with 5 °C increment were derived, and some of the results are shown in Figure 8a. These stress relaxation storage moduli were further approximated by the Ninomiya–Ferry method [21] as:(23)$$E(t)=E^{\prime }(\omega )-0.4E^{\prime \prime }(0.4\omega )+0.014E^{\prime \prime }(10\omega )$$
where ω=1/t, E represents the stress relaxation modulus, $E^{\prime }$ is the storage modulus, and $E^{\prime \prime }$ denotes the loss modulus. It is clear that the stress relaxation storage modulus would have a strong temperature correlation at temperatures neighboring the *T_g_* of the EMC, which was around 100 °C. In addition, the stress relaxation storage modulus near the *T_g_* showed a great time dependence, and at temperatures lower than 50 °C and higher than 200 °C exhibited trivial time and temperature correlations.

Based on the TTS principle, a single master curve could be constructed by shifting these frequency-dependent storage moduli at different temperatures along the time axis, as shown in Figure 8b, where the reference temperature was set to the *T_g_* of the EMC. The master curve could be well fitted by a Prony series equation using 22 Prony elements; the fitted weighting coefficients $c_{k}$ and the relaxation times $\varsigma_{k}$ are listed in Table 1. The corresponding temperature shift factors in logarithmic scale are shown in Figure 9, as a function of temperature. These shift factors were further fitted to the curve using the WLF model and the curve fitting result is also shown in Figure 9. Clearly, there also was a very good fit to these shift factors with the fitted constant values *a*_1_ = 208.9 and *a*_2_ = 1092.0.

### 5.2. Verification of the Effective Models

To demonstrate the feasibility of the TM technique, a fraction of a Cu circuit layer was considered as a test vehicle. The fractional Cu circuit layer was modeled using both detailed FE modeling and the TM technique, and the results are presented in Figure 10. Noticeably, there was a high agreement between them, indicating that the TM technique could not only robustly but also precisely distinguish the Cu circuitries from the PP dielectric or SM material.

The effectiveness of the proposed TM/FEA effective method was verified by comparing the predicted effective orthotropic elastic properties of the fractional Cu circuit layer displayed in Figure 10 with the ROM/analytical estimate and with the FEA-based effective approach. The latter was considered a benchmark model. It was worth mentioning that for simplification, the orthotropic elastic material is simplified as a transversely isotropic elastic material by averaging the effective in-plane elastic moduli ($E_{x}$ and $E_{y}$) and CTEs ($\alpha_{x}$ and $\alpha_{y}$) to be $E_{\bar{x,y}}$ and $\alpha_{\bar{x,y}}$, respectively. The calculated effective properties are shown in Table 2 and Table 3. These two tables illustrate that the effective elastic moduli and CTEs showed a strong temperature dependence, where an elevated temperature would considerably lessen the effective elastic moduli but enlarge the CTEs. Moreover, Table 2 shows that the calculated effective in-plane and out-of-plane elastic moduli by the proposed TM/FEA effective method were much more consistent with those of the FEA-based effective method over the temperature range of 25–260 °C, as compared to the ROM/analytical estimate. On the other hand, the calculated effective in-plane elastic moduli by the ROM/analytical estimate deviated considerably from those of the other two effective approaches across the temperature range. Furthermore, a similar result could be also found for the predicted effective CTEs, as shown in Table 3. Similar to the effective elastic moduli, there is a very pronounced difference in the effective in-plane CTEs between the ROM/analytical estimate and the other two effective approaches.

The orthotropic Cu circuit layer was further approximated as a transversely isotropic material by simply averaging the effective in-plane elastic moduli ($E_{x}$ and $E_{y}$) and CTEs ($\alpha_{x}$ and $\alpha_{y}$) as $E_{\bar{x,y}}$ and $\alpha_{\bar{x,y}}$, respectively. The transversely isotropic material could be further simplified as an isotropic material through an average of the effective in-plane and out-of-plane elastic moduli ($E_{\bar{x,y}}$,$E_{z}$) and CTEs ($\alpha_{\bar{x,y}}$, $\alpha_{z}$) as $E_{\bar{x,y,z}}$ and $\alpha_{\bar{x,y,z}}$, respectively. The feasibility of the three constitutive models, i.e., orthotropic, transversely isotropic and isotropic, was further examined through FEA of the fractional Cu circuit layer shown in Figure 10 when subjected to a temperature load from 25 °C to 260 °C, and the calculated thermal deformations in the *x-*, *y-* and *z-* directions are displayed in Table 4. For comparison, the detailed FEA results, serving as benchmark data, are also listed in the table, which shows that the effective orthotropic model presented the best consistency with the detailed FEA, followed by the transversely isotropic and isotropic models. This result matched with mechanical intuition. Accordingly, the effective orthotropic constitutive model was used in the subsequent warpage process simulation.

### 5.3. Thermal Analysis of the Interposer Bonding Process

The temperature distribution of the FCPoP assembly during the interposer bonding process may not be uniform across the assembly due to the uneven applied process thermal loading, which would cause a more excessive deformation. Thus, prior to conducting the warpage process simulation, the temperature distribution of the FCPoP assembly in natural convection during the interposer bonding process was characterized using a 3D transient heat conduction FEA. The natural convective heat transfer model proposed in [22] and the standard radiative heat transfer model [23] were applied to depict the natural convective and radiative surface heat transfer, respectively. According to the process condition, a preheat temperature was first set on the top and bottom surfaces of the assembly for 4 s, which were 185 °C and 145 °C, respectively, followed by a temperature increase up to 245 °C in 10 s on the top surface. The ambient temperature is 25 °C. The thermal analysis result at the end of the process is demonstrated in Figure 11. It is important to note that the Cu core solder balls were arranged in two to three rows around the periphery of the substrate and interposer. Evidently, the heat was conducted from the top coreless substrate to the bottom core interposer mainly by way of these Cu core solder balls; as a result, the part of the substrate and interposer adjacent to these periphery Cu core solder balls experienced a higher temperature. In addition, there was a significant temperature non-uniformity and gradient across the assembly. The characterized temperature distribution was imposed as a thermal load in the warpage process simulation for a better prediction accuracy.

### 5.4. Warpage Process Simulation

The calculated temperature-dependent effective orthotropic elastic properties of the coreless substrate and core interposer using the TM/FEA effective approach are demonstrated in Table 5 and Table 6, respectively. It was interesting to see that the core interposer was much stiffer than the coreless substrate across the temperature range; On the contrary, the CTEs of the coreless substrate tended to slightly greater than those of the core interposer. The warpage evolution of the FCPoP assembly during the fabrication process is shown in Figure 12a. It can be clearly observed in the figure that the process-induced warpage extensively varied with the process steps, and also showed a significant increase after the die process bonding, underfill curing, and interposer bonding processes. In addition, the maximum warpage occurred after the interposer bonding process; i.e., at around 653.7 μm, rather than after the fabrication process, i.e., at about 82.6 μm. The reason that the interposer bonding process created the maximum warpage was the lack of the EMC helping to resist the shear force caused by the global CTE mismatch between the substrate and interposer.

Figure 12b illustrates the simulated warpages during the increasing temperature process. For comparison, the in-line warpage measurement data are also listed in Figure 12, presented as an average value with a standard deviation (SD) (error bar). It was evident that the simulated warpages over the temperature range of 25–260 °C closely followed the measurement data. The fair difference in warpage between the measurement and simulation could be attributed to the uncertainty in the measured temperature-dependent Young's modulus and CTE of the EMC. Additionally, the increased temperature reduced the warpage probably due to the softening of the EMC when exposed to temperatures greater than the *T_g_*. The simulated and measured warpage contour plots of the FCPoP assembly at step 12 (after the mold cure process) are shown in Figure 13. Once again, there were very consistent results between the simulation and measurement. Table 7 illustrates the simulated and average measured residual warpages at 30 °C and 260 °C together with the smallest and largest values of the measured data shown in the bracket. It is clear that these simulation data fell in the respective ranges of the measured data, and in addition, the results of the process simulation were very comparable to those of the measurement, where the maximum warpage difference between them was only around 5%. Moreover, the residual warpage at 30 °C is nearly double that found at 260 °C, and due to this, the residual warpage at 260 °C was not considered in the subsequent parametric analysis.

### 5.5. Parametric Study

#### 5.5.1. Effects of Component CTEs

The influences of the effective CTEs of the EMC, core interposer and coreless substrate on the warpage of the FCPoP assembly at 30 °C were addressed. The parametric results of the effect of the EMC CTE are presented in Figure 14a. In the parametric analysis, the effective CTE of the EMC nominally varied from −10% to +10%. The figure shows that the EMC CTE had a minor impact on the residual warpages due to the relatively rigid substrate and interposer. Specifically, an increase in the EMC CTE somewhat decreased the residual warpage. This could be due to an increased EMC CTE reducing its local CTE mismatch with the core interposer and coreless substrate, thereby leading to a lessened residual warpage.

The effects of the effective CTEs of the core interposer and coreless substrate are investigated, and the parametric results are also displayed in Figure 14a. Note that the three effective CTEs (*α_x_, α_y_, α_z_*) shown in Table 5 and Table 6 simultaneously underwent a ±10% variation from the original value. The figure demonstrates that the effective CTEs had a significant impact on the residual warpage. Specifically, an increase in the effective interposer CTEs dramatically reduced the warpage, whereas there was a totally opposite trend for the effective substrate CTEs. This was principally due to the effective CTEs of the interposer being smaller than those of the substrate. This suggests that the increase in the interposer CTEs reduced the CTE mismatch with the substrate, thereby leading to a reduced residual warpage. Likewise, the result of the effects of the effective CTEs of the coreless substrate can be also explained in the same way.

#### 5.5.2. Effect of Component Orthotropic Elastic Properties

The effects of the effective orthotropic elastic properties of the core interposer and coreless substrate on the warpage at 30 °C were considered. Similar to the parametric analysis of the effective CTEs, there was a ±10% variation in these 9 independent effective elastic property data ($E_{x}$, $E_{y}$, $E_{z}$, $\upsilon_{xy}$, $\upsilon_{yz}$, $\upsilon_{xz}$, $G_{xy}$, $G_{yz}$ and $G_{xz}$) shown in Table 5 and Table 6. The parametric results are presented in Figure 14b. They indicated that increased effective elastic properties of the interposer and decreased effective elastic properties of the substrate would amplify the residual warpage. This was because the core interposer is stiffer than the coreless substrate due to its possessing greater effective elastic and shear moduli. The growth of the effective elastic shear and elastic moduli of the interposer tended to result in a more excessive shear force resulting from the CTE mismatch between the interposer and substrate, thereby causing a greater warpage. On the other hand, the structural rigidity of the FCPoP assembly increased with the increase of the effective elastic and shear moduli of the substrate, which thus led to a reduced warpage. The results totally differed from the effects of the effective CTEs of the interposer and substrate.

#### 5.5.3. Effect of Component Thickness

The dependence of the warpage at 30 °C on the thickness of the core interposer, coreless substrate and EMC was examined. Similarly, the thickness variation is also ±10% from their original value. Figure 14c illustrates the parametric results, where the residual warpage would increase both with an increasing interposer thickness and with a decreasing substrate thickness. The results were very consistent with the effects of the elastic properties of the interposer and substrate, and the explanation for this is the same as stated in the pervious section. In the parametric analysis of the thickness effect of the EMC, parametrizing the EMC thickness would, in the meantime, change the height of the CCSBs. Before conducting the parametric study of the influence of the EMC thickness, the height effect of the CCSBs in the assembly without an EMC was first explored. The parametric study, which is not presented here due to limited space, suggested that the CCSBs’ height had little impact for the residual warpage. As a result, the parametric results on the thickness effect of the EMC was barely affected by the CCSBs’ height. The dependence of the residual warpage on the EMC thickness is also presented in Figure 14c, which shows that the residual warpage significantly decreased with the EMC thickness. This can be attributed to the structural stiffness of the assembly substantially increasing with the EMC thickness, thus resulting in a less residual warpage.

## 6. Conclusions

This research successfully conducted an effective and robust prediction of the warpage performance of an FCPoP assembly during the fabrication process through the proposed process simulation framework. In this framework, the temperature-dependence of the elastic properties of the components, as well as the viscoelastic behavior and chemical shrinkage of the EMC were taken into account in this investigation. The temperature-dependent elastic properties and viscoelastic properties were experimentally characterized. In order to improve the computational and modeling efficiency while also preserving good prediction accuracy, a novel effective approach; i.e., the TM/FEA effective method, was introduced to assess the effective elastic properties of the orthotropic coreless substrate and core interposer. The effectiveness of the proposed effective method and the proposed process simulation framework were extensively validated. Finally, a parametric analysis was performed to investigate the dependence of the process-induced warpage on some geometric and material parameters.
The ECAD TM technique was a very effective and robust way to precisely recognize the Cu circuitries in the PP dielectric or SM material.The DMA results indicated that the storage modulus of the EMC showed great time and temperature dependence particularly at temperatures near its *T_g_*.Both the predicted effective elastic moduli and CTEs of the substrate and interposer turned out to have a negative and a positive temperature coefficient, respectively.The orthotropic constitutive assumption was shown to provide the most accurate prediction of the thermal deformations of the substrate and interposer, as compared to the transversely isotropic and isotropic ones.The thermal analysis results showed that there was a significant temperature non-uniformity across the assembly during the interposer bonding process, which could potentially affect the process-induced warpage.The proposed TM/FEA effective method and proposed process simulation framework were found to be very effective in predicting the effective elastic properties of the substrate and interposer and the process-induced warpage of the FCPoP assembly, respectively.The process-induced warpage of the FCPoP assembly experienced a dramatic change over the process steps, and more importantly, the maximum warpage occurred after the interposer bonding process rather than the end of the fabrication process. In addition, the warpage at 30 °C was roughly twice that of 260 °C.The warpage decreased with temperature during the increasing temperature process, probably because the EMC material became softened at temperatures greater than the *T_g_*.Among the parameters considered in the parametric analysis, the substrate CTE had the greatest influence on the warpage at 30 °C, followed by the interposer CTE and the EMC thickness; moreover, a smaller substrate CTE, a larger interposer CTE and a thicker EMC brought about a reduced warpage.

## Figures and Tables

**Figure 1 materials-14-04816-f001:**
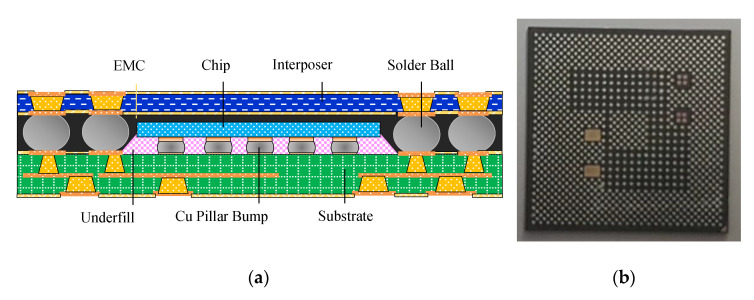
The FCPoP assembly: (**a**) cross-sectional view; (**b**) prototype.

**Figure 2 materials-14-04816-f002:**
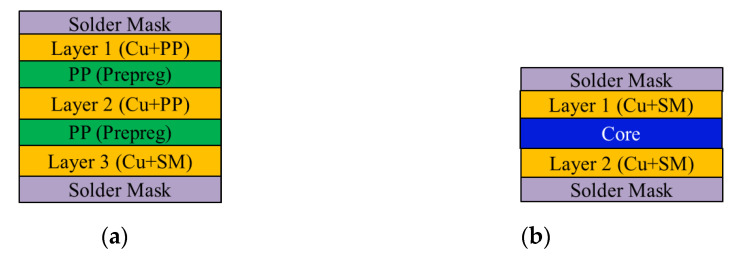
Schematic cross-sectional view: (**a**) top coreless substrate; (**b**) bottom core interposer.

**Figure 3 materials-14-04816-f003:**
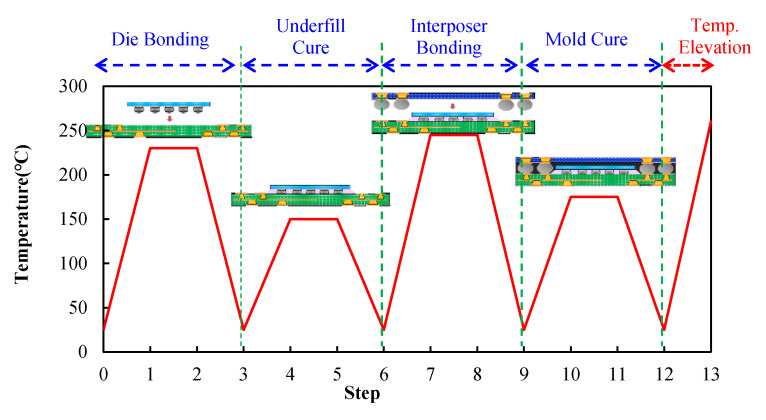
Fabrication process steps with temperature loads.

**Figure 4 materials-14-04816-f004:**
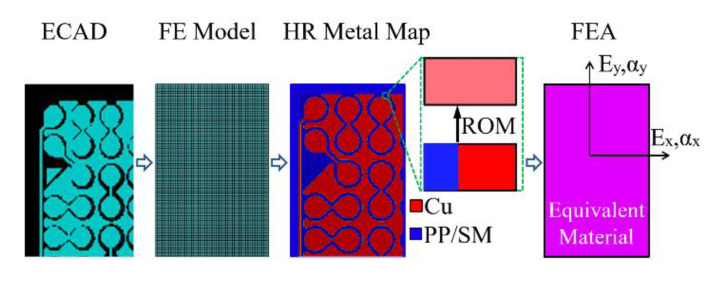
A flowchart of ECAD TM method.

**Figure 5 materials-14-04816-f005:**
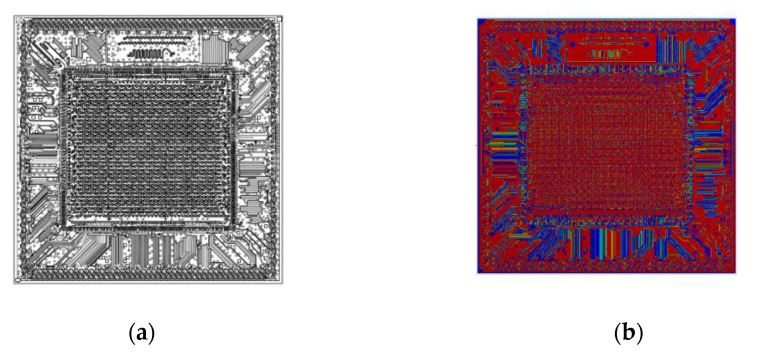
A Cu circuit layer in a substrate: (**a**) Cu circuitry pattern; (**b**) Approximate FE model.

**Figure 6 materials-14-04816-f006:**
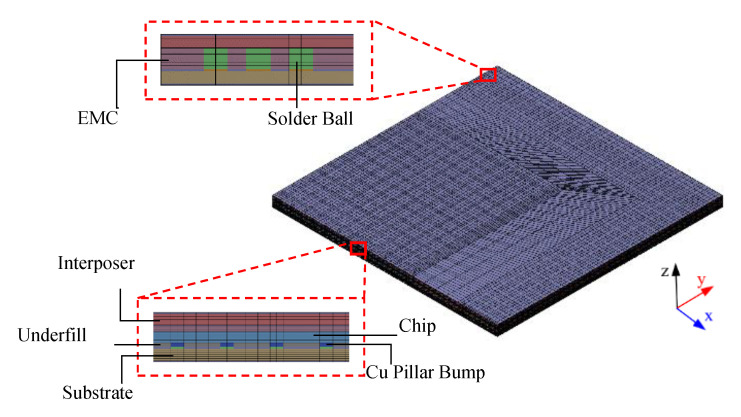
3D FE model of the FCPoP assembly.

**Figure 7 materials-14-04816-f007:**
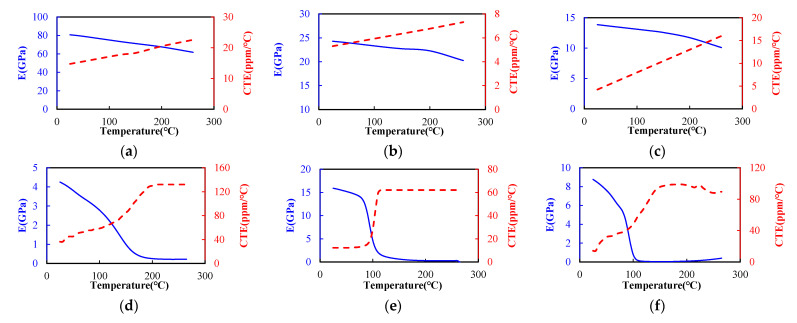
Temperature-dependent Young’s modulus (solid line) and CTE (dashed line) of the components: (**a**) Cu; (**b**) core material; (**c**) PP; (**d**) solder mask; (**e**) EMC; (**f**) underfill.

**Figure 8 materials-14-04816-f008:**
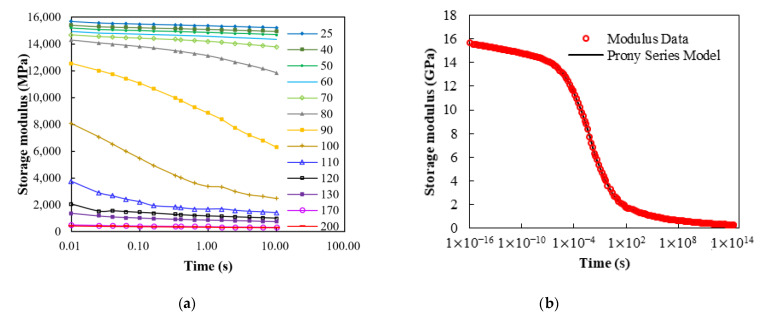
(**a**) Stress relaxation storage moduli at different isothermal temperatures; (**b**) Construction of single master curve and its Prony series representation.

**Figure 9 materials-14-04816-f009:**
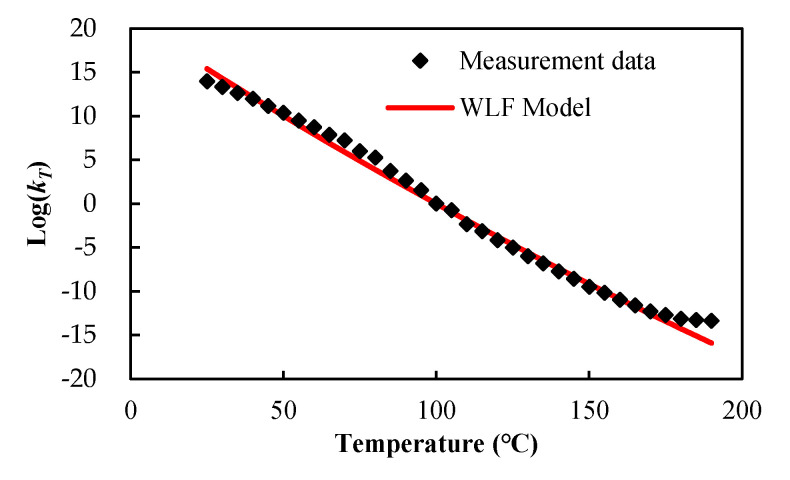
Temperature shift factors and WLF curve fitting.

**Figure 10 materials-14-04816-f010:**
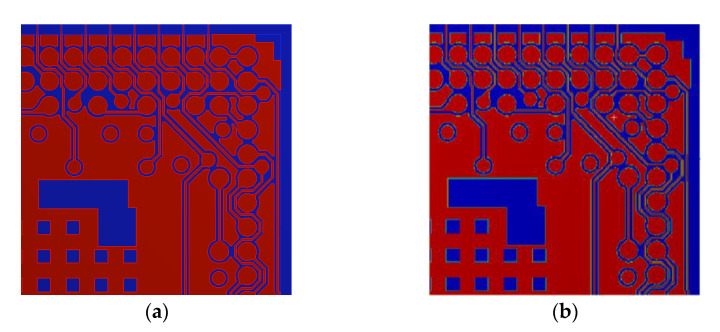
Constructed FE models of a fractional Cu circuit layer using: (**a**) detailed FE modeling; (**b**) TM method.

**Figure 11 materials-14-04816-f011:**
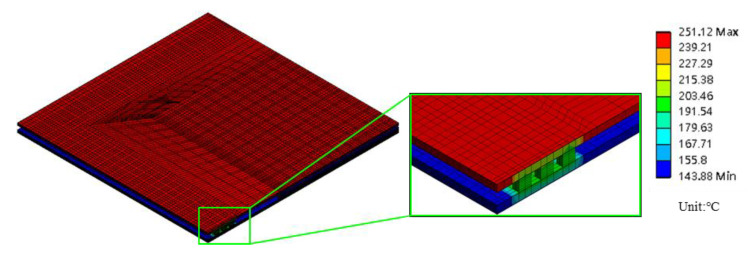
Temperature distribution of the assembly during interposer bonding process.

**Figure 12 materials-14-04816-f012:**
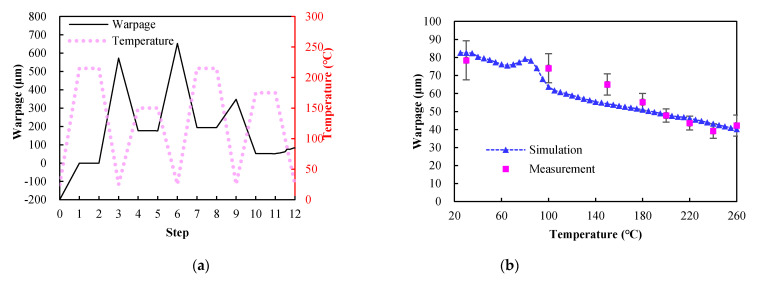
Warpage evolution: (**a**) from die bond to mold cure process; (**b**) during increasing temperature process.

**Figure 13 materials-14-04816-f013:**
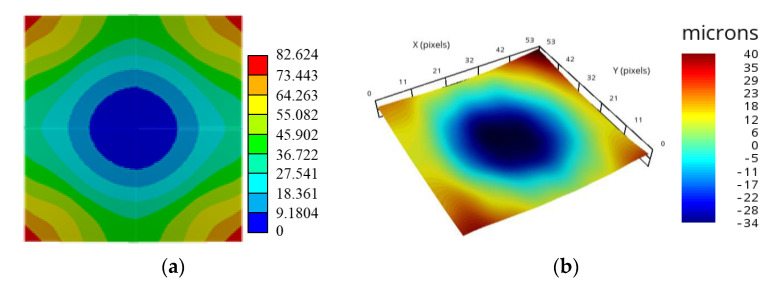
Comparison of simulated and experimental warpages after the mold cure process; (**a**) simulation; (**b**) experiment.

**Figure 14 materials-14-04816-f014:**
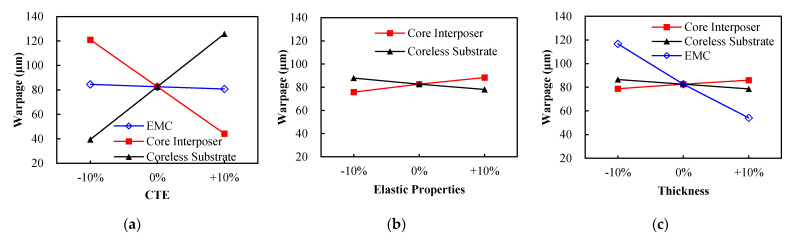
Effects of component material and geometric parameters: (**a**) CTE; (**b**) orthotropic elastic properties; (**c**) thickness.

**Table 1 materials-14-04816-t001:** Fitted values of weighting coefficients and relaxation times.

k	$\varsigma_{k}$	$c_{k}$	k	$\varsigma_{k}$	$c_{k}$	k	$\varsigma_{k}$	$c_{k}$	k	$\varsigma_{k}$	$c_{k}$
1	1.0 × 10^−16^	0.0241	7	1.0 × 10^−6^	0.0259	13	1.0 × 10^0^	0.0743	19	1.0 × 10^8^	0.0092
2	1.0 × 10^−14^	0.0128	8	1.0 × 10^−5^	0.0465	14	1.0 × 10^1^	0.0474	20	1.0 × 10^10^	0.0073
3	1.0 × 10^−12^	0.0139	9	1.0 × 10^−4^	0.0796	15	1.0 × 10^2^	0.0233	21	1.0 × 10^12^	0.0052
4	1.0 × 10^−10^	0.0162	10	1.0 × 10^−3^	0.0894	16	1.0 × 10^3^	0.0089	22	1.0 × 10^14^	0.0035
5	1.0 × 10^−8^	0.0169	11	1.0 × 10^−2^	0.1411	17	1.0 × 10^4^	0.0208			
6	1.0 × 10^−7^	0.0122	12	1.0 × 10^−1^	0.0948	18	1.0 × 10^6^	0.0150			

**Table 2 materials-14-04816-t002:** Comparison of calculated effective elastic moduli (MPa) using three different approaches.

T (°C)	FEA-Based		TM/FEA				ROM/Analytical			
	$E_{\bar{x,y}}$	$E_{z}$	$E_{\bar{x,y}}$	Diff. (%)	$E_{z}$	Diff. (%)	$E_{\bar{x,y}}$	Diff. (%)	$E_{z}$	Diff. (%)
25	40,317	56,033	40,690	0.9	55,377	−1.2	30,951	−23.2	58,444	4.3
50	39,536	54,915	39,899	0.9	54,283	−1.2	30,366	−23.2	57,274	4.3
120	37,088	51,125	37,397	0.8	50,581	−1.1	28,650	−22.8	53,272	4.2
150	36,043	49,614	36,337	0.8	49,102	−1.0	27,874	−22.7	51,688	4.2
200	33,914	46,975	34,202	0.8	46,510	−1.0	26,102	−23.0	48,976	4.3
260	30,122	42,614	30,422	1.0	42,219	−0.9	22,817	−24.3	44,546	4.5

**Table 3 materials-14-04816-t003:** Comparison of the calculated effective CTEs using three different approaches.

T (°C)	FEA-Based		TM /FEA				ROM/Analytical			
	$\alpha_{\bar{x,y}}$	$\alpha_{z}$	$\alpha_{\bar{x,y}}$	Diff. (%)	$\alpha_{z}$	Diff. (%)	$\alpha_{\bar{x,y}}$	Diff. (%)	$\alpha_{z}$	Diff. (%)
25	11.99	13.76	11.90	−0.8	13.66	−0.7	10.47	−12.6	13.87	0.8
50	12.92	14.62	12.84	−0.6	14.53	−0.6	11.47	−11.2	14.73	0.7
120	15.43	16.90	15.36	−0.5	16.81	−0.5	14.19	−8.1	17.00	0.6
150	16.27	17.58	16.21	−0.4	17.51	−0.4	15.15	−6.9	17.67	0.5
200	18.55	19.82	18.49	−0.3	19.76	−0.3	17.47	−5.8	19.90	0.4
260	20.90	22.03	20.86	−0.2	21.99	−0.2	19.92	−4.7	22.10	0.3

**Table 4 materials-14-04816-t004:** Comparison of calculated thermal deformations among three different constitutive models.

	U_x_ (mm)	Diff. (%)	U_y_ (mm)	Diff. (%)	U_z_ (mm)	Diff. (%)
Detailed FEA	1.14 × 10^−2^	-	1.17 × 10^−2^	-	4.56 × 10^−4^	-
Orthotropic	1.13 × 10^−2^	−0.47	1.17 × 10^−2^	−0.44%	4.58 × 10^−4^	0.52%
Transversely Isotropic	1.15 × 10^−2^	1.04	1.15 × 10^−2^	−1.90%	4.22 × 10^−4^	−7.52%
Isotropic	1.18 × 10^−2^	3.96	1.18 × 10^−2^	0.94%	4.34 × 10^−4^	−4.84%

**Table 5 materials-14-04816-t005:** Temperature-dependent effective orthotropic elastic properties of the coreless substrate (unit: MPa, °C/ppm).

T (°C)	*E_x_*	*E_y_*	*E_z_*	*υ_xy_*	*υ_yz_*	*υ_xz_*	*G_xy_*	*G_yz_*	*G_xz_*	*α_x_*	*α_y_*	*α_z_*
25	27,279	25,603	20,012	0.3	0.3	0.3	3195	3196	3244	12.6	12.1	6.2
50	26,709	25,039	19,631	0.3	0.3	0.3	3115	3132	3182	13.7	13.2	7.3
120	24,871	23,212	18,531	0.3	0.3	0.3	2846	2944	3001	16.4	16.0	10.7
150	24,011	22,277	18,003	0.3	0.3	0.3	2669	2841	2912	17.4	16.9	12.2
200	22,463	20,721	16,796	0.3	0.3	0.3	2411	2628	2713	19.9	19.2	14.9
260	19,991	18,347	14,604	0.3	0.3	0.3	2122	2285	2358	22.4	21.8	17.7

**Table 6 materials-14-04816-t006:** Temperature-dependent effective orthotropic elastic properties of core interposer (unit: MPa, °C/ppm).

T (°C)	*E_x_*	*E_y_*	*E_z_*	*υ_xy_*	*υ_yz_*	*υ_xz_*	*G_xy_*	*G_yz_*	*G_xz_*	*α_x_*	*α_y_*	*α_z_*
25	37,149	37,205	33,475	0.3	0.3	0.3	4636	6028	6023	12.7	12.4	7.2
50	36,423	36,479	32,967	0.3	0.3	0.3	4533	5934	5929	13.4	13.3	7.7
120	33,884	33,940	31,178	0.3	0.3	0.3	4158	5611	5599	14.9	14.9	9.0
150	32,686	32,745	30,183	0.3	0.3	0.3	3930	5477	5445	15.3	15.2	9.6
200	30,904	30,966	27,634	0.3	0.3	0.3	3533	5295	5168	16.8	16.7	11.0
260	28,160	28,218	25,099	0.3	0.3	0.3	3212	4822	4703	18.3	18.2	12.1

**Table 7 materials-14-04816-t007:** Warpage comparison between the simulation and measurement.

Method	Warpage (μm)	
	30 °C	260 °C
Simulation	82.6	40.2
Measurement	78.4 (60,89)	42.2 (32,49)

## Data Availability

Data sharing not applicable.

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
