# Peer review of "Theoretical and Experimental Investigation of Warpage Evolution of Flip Chip Package on Packaging during Fabrication"

_materials, 2021, doi:10.3390/ma14174816_

Round 1

Reviewer 1 Report

The Manuscript ID: materials-1315797 was good presented.

This paper attempts to investigate the warpage behavior of a 
flip chip package-on-package  (FCPoP) assembly during fabrication process. A process simulation framework that integrates thermal and mechanical finite element analysis (FEA), effective modeling and ANSYS element deathbirth technique is introduced for effectively predicting the process-induced warpage. 

Please publish in the version after minor revision.

On line 353, "Fig. 10" is intended to be "Fig. 11".

In line 354 it says "from 25-260 ° C" is meant to be "from 25 ° C to 260 ° C"

In Figure 11, please enter the unit "° C". 

On line 378, "Fig. 11" is intended to be "Fig. 12".

Please check the numbering Figures in the text is incorrect.

There are only 20 References, are there no publications on this topic from elsevier or scopus?

Author Response

Dear Reviewer: Please see the attachment. Thanks.

Reviewer 2 Report

The article shows a proper correlation between the theoretical models and the numerical modelling that has been applied. However, I have several important comments:

  1. Figure 8 presents Measurement Data of the Storage modulus vs. Time (sec.) going up to E+14. I think there is an error on the time scale, since, according to the current version, the measurements phase lasted for many years.

Furthermore, I think in the same chart you wanted to write Prony series Model, not PORNY.

  1. According to Figure 11, there is no explanation regarding the temperature distribution between the left and the right parts (on the right part-the thermal exchange is very clear) of the assembly. The thermal simulation should be followed up by detailed structural simulation in order to pass to the following step, namely warpage.
  2. The deployed simulation models and the corresponding phenomena are well detailed, but I suggest the authors to add more details on the specific algorithm used for transitioning EMC region and underfill area (please see Figure 6 -the dotted marked areas).
  3. Which are the benefits of the proposed process simulation framework as compare to SotA methods? In the Introduction section, I saw no reference/comment to other methods used for investigation of Warpage Evolution of Flip Chip Package on Packaging During Fabrication. A comparison with current methods would highlight the novelty of your article.

Author Response

(The authors gave the same response as above.)

Reviewer 3 Report

The paper provides a high quality research but there are some questions:

  • In the relation (4) what is the significance of c?
  • How the results fit into the results from the scientific literature?

Author Response

(The authors gave the same response as above.)

Round 2

Reviewer 2 Report

I can accept your answers, but I do not agree with the maintenance of Figure 8-b (Measurement Data). As the authors themselves stated, the x-axis time scale spanning from 1.0E-16 to 1.0E + 14 is not a real measurement time. I propose to reformulate the term "Measurement Data".

Author Response

(The authors gave the same response as above.)
